# The limited reach of fake news on Twitter during 2019 European elections

**Matteo Cinelli**[1], **Stefano Cresci** [2], **Alessandro Galeazzi** [3]*, **Walter Quattrociocchi**[4], **Maurizio Tesconi**[2]

**1** ISC CNR, Rome, Italy, **2** IIT CNR, Pisa, Italy, **3** University of Brescia, Brescia, Italy, **4** Ca'Foscari Univerisity of Venice, Venice, Italy

* a.galeazzi002@unibs.it

## Abstract

The advent of social media changed the way we consume content, favoring a disintermediated access to, and production of information. This scenario has been matter of critical discussion about its impact on society, magnified in the case of the Arab Springs or heavily criticized during Brexit and the 2016 U.S. elections. In this work we explore information consumption on Twitter during the 2019 European Parliament electoral campaign by analyzing the interaction patterns of official news outlets, disinformation outlets, politicians, people from the showbiz and many others. We extensively explore interactions among different classes of accounts in the months preceding the elections, held between 23rd and 26th of May, 2019. We collected almost 400,000 tweets posted by 863 accounts having different roles in the public society. Through a thorough quantitative analysis we investigate the information flow among them, also exploiting geolocalized information. Accounts show the tendency to confine their interaction within the same class and the debate rarely crosses national borders. Moreover, we do not find evidence of an organized network of accounts aimed at spreading disinformation. Instead, disinformation outlets are largely ignored by the other actors and hence play a peripheral role in online political discussions.

## Introduction

The wide diffusion of online social media platforms such as Facebook and Twitter raised concerns about the quality of the information accessed by users and about the way in which users interact with each other [1–8]. Recently, the chairman of Twitter announced that political advertisements will be banned from Twitter soon, claiming that our democratic system are not prepared to deal with the negative consequences brought by the power and influence of online advertising campaigns [9]. In this context, a wide body of scientific literature focused on the influence and on the impact of disinformation and automation (i.e., social bots) on political elections [10–18]. In [10] the authors studied the impact of fake news on the 2016 US Presidential elections, finding that users sensitivity to misinformation is linked to their political leaning. In [11] is highlighted that fake news consumption is limited to a very small fraction of users with well defined characteristics (middle aged, conservative leaning and strongly

**Data Availability Statement:** All data are currently available at: https://github.com/cinhelli/Limited-Reach-Fake-News-Twitter-2019-EU-Elections.

**Funding:** MC acknowledges the support from CNR-PNR National Project DFM.AD004.027 289

"Crisis-Lab" and P0000326 project AMOFI (Analysis and Models OF social medIa).

**Competing interests:** The authors have declared that no competing interests exist.

engaged with political news). Authors of [12] studied the spreading of news on Twitter in a 10 years time span and found that, although false news spread faster and broader than true news, social bots boost false and true news diffusion at the same rate. The pervasive role of social bots in the spread of disinformation was instead reported in [19] for financial discussions, where as much as 71% of users discussing hot US stocks were found to be bots. The effects of fake and unsubstantiated news affected also the outcome of other important events at international level. For instance, the evolution of the Brexit debate on Facebook has been addressed in [20] where evidence about the effects of echo chambers, confirmation bias in news consumption and clustering are underlined. Nevertheless, as stated in [21], the conclusions of these and other studies are partially conflicting. This conflict can be the result of the differences in the definitions of fake news or misinformation adopted by different authors, that have somewhat contributed in switching the attention from the identification of fake news to the definition itself.

In particular, authors in [1] and [22] focused their attention on the process that can boost the spreading of information over social media. In these works, it is highlighted that phenomena such as selective exposure, confirmation bias and the presence of echo chambers play a pivotal role in information diffusion and are able to shape the content diet of users. Given the central role of echo chambers in the diffusion process, authors of [23] propose a methodology based on users polarization for the early identification of topics that could be used for creating misinformation. However, in [24] it is stressed that the phenomenon of echo chambers can drive the spreading of misinformation and that apparently there are no simple ways to limit this problem.

Considering the increasing attention paid to the influence of social media on the evolution of the political debate, it becomes of primary interest to understand, at a fast pace, how different actors participate in the online debate. Such concerns are renewed in the view of the upcoming US Presidential Election of November 2020 or the future national elections in EU countries.

The goal of our work is to characterize the information flow among different actors that took place in the run up to the last European Parliament elections held between the 23rd and 26th of May, 2019. According to the European legislation, every 5 years all the country members of the EU have to hold elections to renovate their members at the European Parliament. The election can be held in a temporal window of few days and every state can decide in which days to hold the voting procedure. During the electoral campaign, concerns about the impact of fake news on the upcoming European election were risen by several news outlets [25] and misinformation have been monitored, also thanks to the effort of NGOs, in different platforms [26]. The EU itself started a joint and coordinated action on misinformation mitigation [27]. Based on what happened during Brexit and the US 2016 election, also EU leaders encouraged the adoption of measures at the European level to counteract the diffusion and impact of Fake News [28]. Additional evidence of the potential impact of misinformation during European Election motivated studies at national level such as [29]. Starting from these premises, our study aims to assess the reach of fake news during European Elections. In this context, we characterize the public debate on Twitter in the months before the elections. In particular, we aim at understanding which role was played by users that have different positions in public society, including disinformation outlets and popular actors either directly or indirectly related to politics, to obtain a wide view of the process. Through a thorough quantitative analysis on a dataset of 399,982 tweets posted by 863 accounts in the three months before the elections, we first analyze the information flow from a geographical point of view and then we characterize the interactions among different classes of actors. Finally, we compare the impact of disinformation-related accounts with respect to all others. We find that all classes, except official news

outlets, have a strong tendency towards intra-class interaction and that the debate rarely cross the national borders. Moreover, disinformation spreaders have a marginal role in the information exchange and are ignored by other actors, despite their repeated attempts to join the conversation. Although the maximum outreach of fake news accounts is lower than that of other categories, when we take into account comparable levels of popularity we observe an outreach for disinformation that is larger than that of traditional outlets and comparable to that of politicians. Such evidence demonstrates that disinformation outlets have a rather active followers base. However, the lack of interactions between fake news accounts and others demonstrated that their user base is confined to a peripheral portion of the network, suggesting that the countermeasures taken by Twitter, such as suspension or ban of suspicious accounts, might have been effective in keeping the Twittersphere clean.

## Results and discussion

By exploiting Twitter APIs, we collected data from the Twitter timelines of 863 users. This resulted in the acquisition of 399,982 tweets shared between February 28 and May 22, 2019. The 863 users in our dataset are classified into 8 categories, based on their roles in the society. In detail, we have categories encompassing trusted news outlets (labeled `official`), politicians, disinformation outlets (`fake`), show business personalities (`showbiz`), official accounts of social media platforms, sport personalities, famous brands (`trademarks`), and other VIPs. By leveraging information contained in tweets and users metadata that we collected, we also computed the interactions between all the accounts of our dataset and we geolocated Twitter users, whenever possible. A detailed view of our Twitter dataset is summarized in Table 1 while additional information is available in the "Materials and Methods" Section. By leveraging account interactions, we built a directed graph $G = (V, E)$ where each node $v_i \in V$ corresponds to a Twitter account and each link $e_i = (v_A, v_B) \in E$ from node $v_A$ to node $v_B$ exists (i.e., Ⓐ→Ⓑ) if and only if $v_A$ interacted with $v_B$ in one of the following ways: (i) $v_A$ **retweeted** $v_B$; (ii) $v_A$ **replied** to $v_B$; (iii) $v_A$ **mentioned** $v_B$ in a tweet; (iv) $v_A$ **tweeted a link to an article** that mentioned $v_B$. We refer to the last type of interaction as *indirect*—whereas all others are *direct*—since Web links do not point directly to Twitter accounts, but rather point to Web pages outside Twitter that, in turn, mention accounts in our dataset. Our rich interaction network is thus representative of the information flow across different actors, including disinformation outlets, and several countries involved in the 2019 European Parliament elections.

We first characterize the geographical composition of our dataset. As shown in Fig 1, our dataset is mainly made up of accounts located in the EU and the US. However, a small fraction

**Table 1. Dataset summary.**

| | class | users | tweets | interactions | | | |
|---|---|---|---|---|---|---|---|
| | | | | *retweets* | *replies* | *mentions* | *articles* |
| ⬤ | fake | 45 | 24,331 | 4,375 | 2,640 | 12,927 | 4,389 |
| ⬤ | official | 333 | 207,171 | 49,515 | 9,966 | 99,595 | 48,095 |
| ⬤ | politicians | 328 | 88,627 | 23,188 | 5,603 | 57,512 | 2,324 |
| ⬤ | showbiz | 98 | 29,873 | 5,414 | 2,838 | 21,475 | 146 |
| ⬤ | social media | 8 | 8,824 | 402 | 3,901 | 4,499 | 22 |
| ⬤ | sport | 37 | 33,616 | 6,057 | 2,059 | 25,490 | 10 |
| ⬤ | trademarks | 6 | 4,289 | 207 | 1,789 | 2,293 | 0 |
| ⬤ | VIPs | 11 | 3,251 | 192 | 812 | 2,238 | 9 |
| | total | 863 | 399,982 | 89,350 | 29,608 | 226,029 | 54,995 |

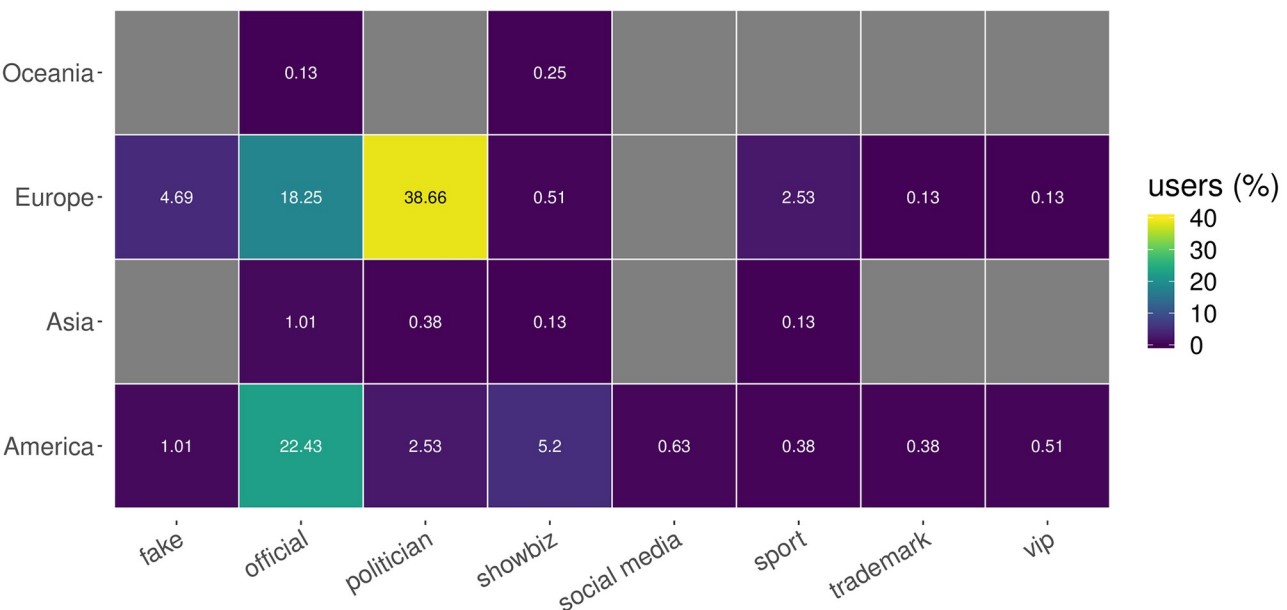

**Fig 1. Heatmap showing the distribution of users interacting with the different actor classes, per geographic area.**

of accounts belong to other parts of the world. This is due to the fact that we integrated our initial set of accounts with a subset of popular accounts (more than 1M followers) that interacted with them. Notably, only a small fraction of accounts belong to non EU/US places. This may be a first signal that the interactions rarely cross national borders. Indeed, the top panel of Fig 2 shows the geographic distribution of user interactions on a world map, while the bottom panel represents the information as a chord diagram where interactions are grouped by actor class and by country. The top panel highlights that the vast majority of interactions (65%) is initiated by official accounts (green links) and that a considerable number of links between the US and the EU (10%) exists. The chord diagram of Fig 2 provides more details about countries and classes, confirming that the biggest contribution to the debate is provided by official accounts, followed by politicians. However, it is noticeable that most of the links start and end in the same country, while the center of the chord diagram is almost empty, implying that the debate rarely crossed national borders. The only relevant exception is represented by official news outlets that tend to cite politicians from other countries (11% of all links). This is particularly clear for the UK, where a relevant fraction of links coming from official accounts point to US politicians (36% of all links from UK news outlets)—that is, UK news outlets tweet about US politicians quite often. All other groups tend to refer only to accounts from the same country and often also of the same type. Although a precise assessment of the causes of this phenomenon is beyond the scope of this paper, we provide further details and briefly discuss the possible impact of language barriers in the SI.

In order to understand how accounts of the same type interact among themselves, we induced subgraphs based on node categories hence obtaining one subgraph for each category. Fig 3 shows the subgraphs plotted in a world map, for the four biggest classes of actors: fake (panel **A**), politicians (panel **B**), official (panel **C**), and showbiz (panel **D**). We note that only subgraphs related to official news outlets, politicians and showbiz accounts are well connected. Indeed, the proportion of nodes belonging to the largest connected component is respectively 66%, 91% and 84% of the total number of nodes. On the contrary, the graph related to disinformation news outlets (panel **A**) comprises mostly isolated nodes. In the case

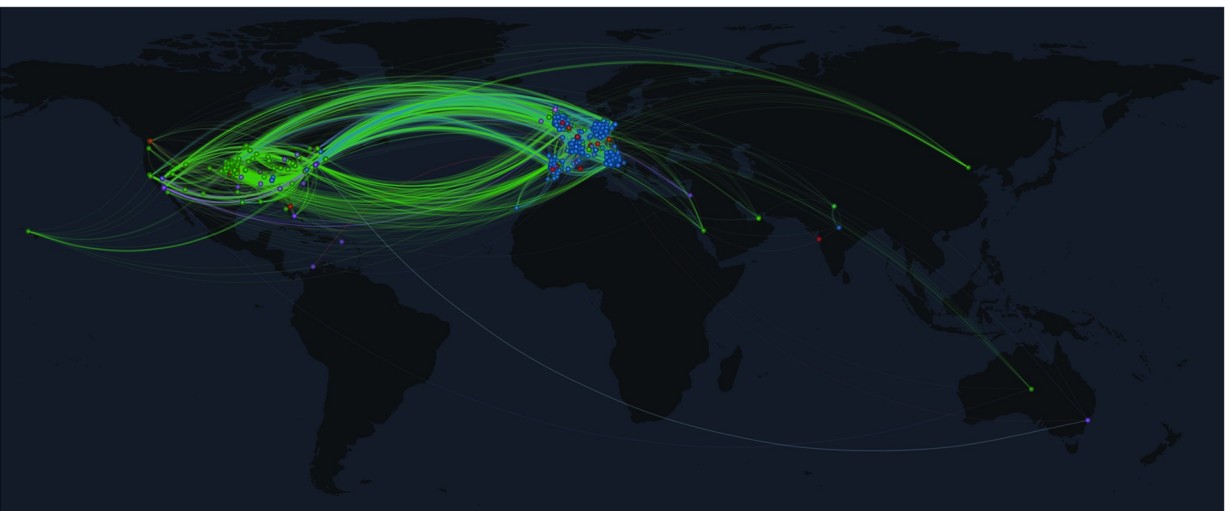

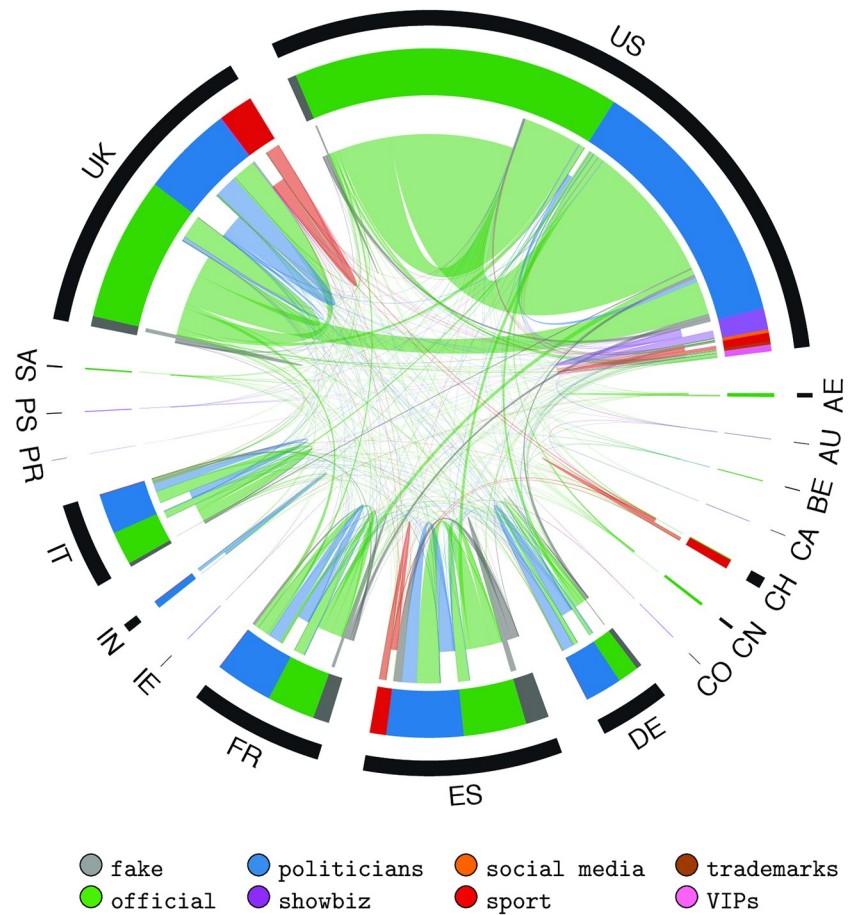

**Fig 2. Node-link diagram showing the geographic representation of information flows (top) and Chord diagram showing class interactions grouped by country (bottom) during EU elections.** Loops are taken into account only in the chord diagram, that highlights the tendency of accounts to interact mainly with users in the same nation and often also in the same class.

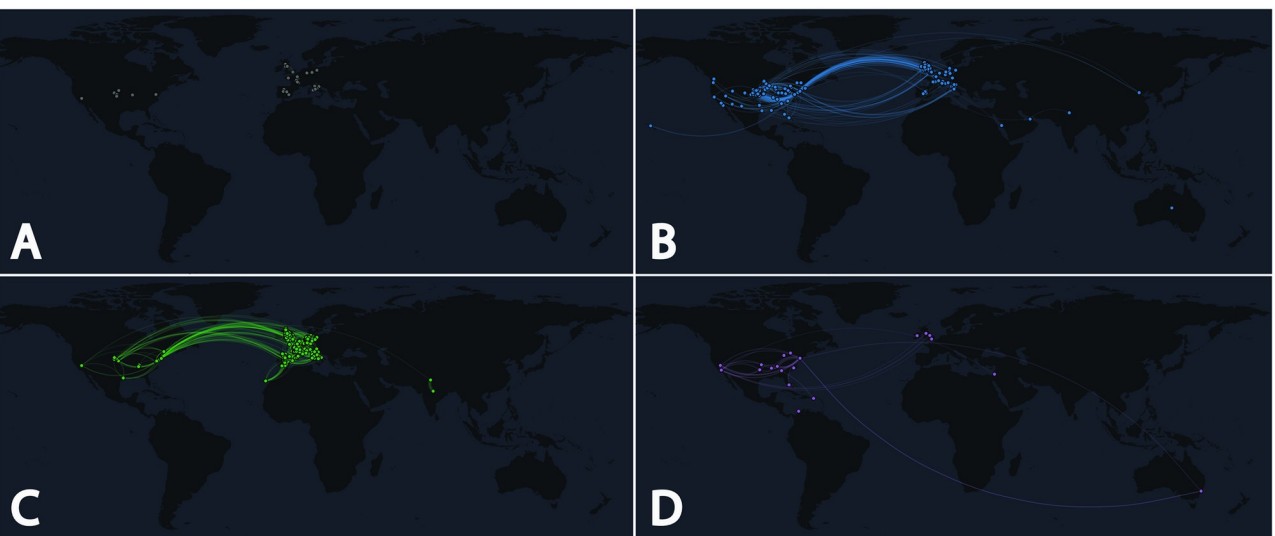

**Fig 3.** Geographic representation of intra-class interactions for the four biggest classes of actors: fake (panel **A**), politicians (panel **B**), official (panel **C**), and showbiz (panel **D**). Notably, panel **A** has only one link between two nodes in the UK, while all other panels exhibit a large number of interactions. For clarity, self-loops are omitted.

of disinformation news outlets the nodes belonging to the largest connected component are about 9% of the total number of nodes. Such evidence suggests that Twitter accounts related to disinformation outlets rarely dialogue with their peers, but rather they prefer to interact with other types of actors. Furthermore, comparing Fig 3 with the chord diagram of Fig 2, we can infer that outlets labelled as fake also display a tendency towards self-mentioning. Instead, politicians and showbiz accounts show a relevant percentage of interactions with others from the same class (respectively 42% and 22%, without considering self interactions) while official news outlets interact mainly with other classes (71% of total links amount). Although there are some similarities in the statistics of fake and official outlets—that is, both try to interact with other classes—only official accounts catch the attention of other actors, while fake outlets are most of the times ignored. To clarify the way in which different actors participated in the debate, we also analyzed the proportion of incoming and outgoing links by class. Results are shown in Fig 4. In the first row all types of interactions were considered (i.e., both *direct* and *indirect*), while in the second one only *direct* interactions (retweets, replies, mentions) were taken into account. Some differences arise when comparing all outgoing links with *direct* outgoing links (left-hand side of Fig 4), in particular with regards to the classes fake and official. When all kinds of interactions (*direct*+*indirect*) are taken into account, we note an increment in the fraction of outgoing links that point to politicians (blue-colored bar, +57% and +51% respectively) for both classes. In other words, the classes labelled as fake and official interact with politicians mainly through external resources. These could be news articles mentioning politicians, that are linked and shared in Twitter.

The proportions of incoming links are shown in the right-hand side of Fig 4. The most relevant difference between *direct* and *indirect* links concerns the category of politicians. In fact, there is an increase of links coming from the official and fake classes (+49% and +5%) that is in accordance with the differences found in the case of outgoing links. Again, we notice that accounts, except for official ones, display the tendency to interact mainly within their own classes, and this is even more evident when only *direct* links are taken into consideration. Finally, by analyzing the behavior of the official and fake classes, we noticed that both of them mainly

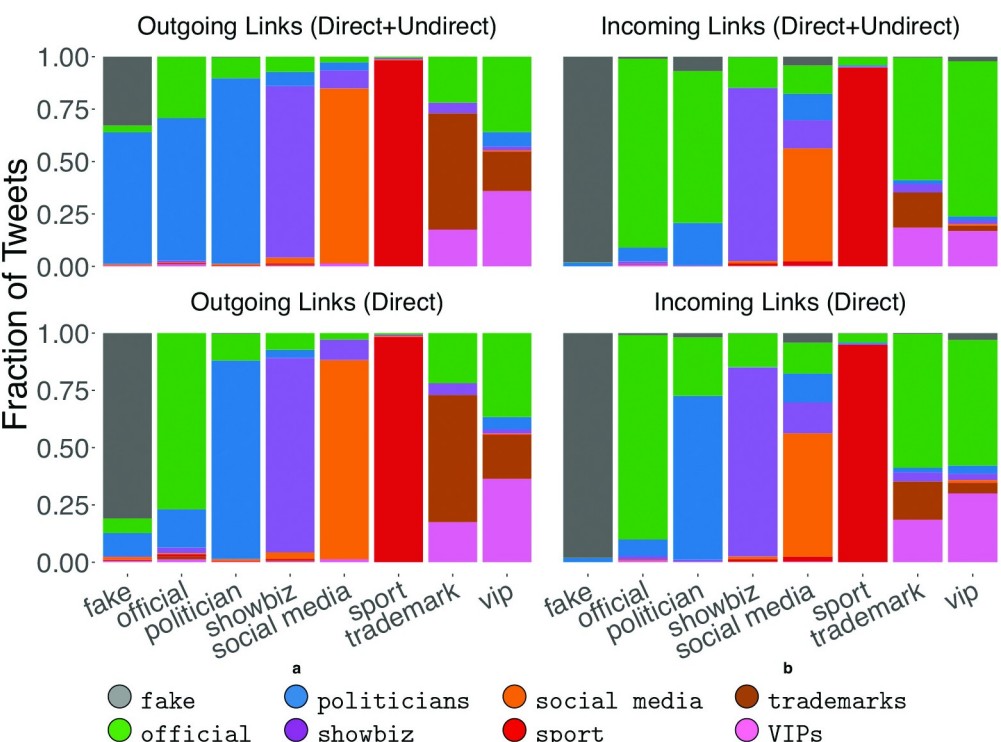

**Fig 4. Outgoing and incoming links by class.** The top row accounts for all types of interactions, the bottom one only considers *direct* interaction (i.e., replies, retweets, mentions). For this analysis self-loops are considered, which explains the tendency of all classes towards self-interaction.

refer to politicians when external sources are taken into account. However, politicians mostly interact among themselves and only a small fraction (9%) of their outgoing links are directed to official accounts, with disinformation outlets being substantially ignored. Indeed, we measure the number of nodes connected by reciprocal *direct* links (i.e. A and B are connected in both direction with a link representing a mention, a retweet or a reply) among the classes. We found that fake news accounts, news outlets and politicians reach progressively higher reciprocity scores especially within their own classes. The average percentage of nodes connected by reciprocated links in the same class is $\mu = 23.4\%$ and only 9% of fake news accounts are reciprocally interconnected. Moreover, fake news accounts exhibit a behavior that differs from other classes when the percentage of nodes connected by reciprocated inter-class links is taken into account: while the average percentage is $\mu = 5.5\%$, fake news accounts do not display mutual connections with any other class. Such evidence, combined with the information conveyed in Figs 2 and 4, suggests that disinformation outlets try to fit in the political debate, but they are essentially ignored by mainstream news sources, by politicians, and also by the other classes of actors. Interestingly, the behavior of fake news accounts is akin to that of automated accounts as shown by the authors of [30] during the Catalan Referendum: both fake news and automated accounts tend to target popular users in order to increase their relevance and impact in the public debate. Our previous finding indicates that Twitter accounts related to disinformation outlets did not seem to be able to enter the main electoral debate. However, despite not attracting interest from the main actors involved in the debate, they could still have had an impact on the general audience. To investigate this issue we study the engagement obtained by the different classes of actors. In particular, each actor produces tweets, and each

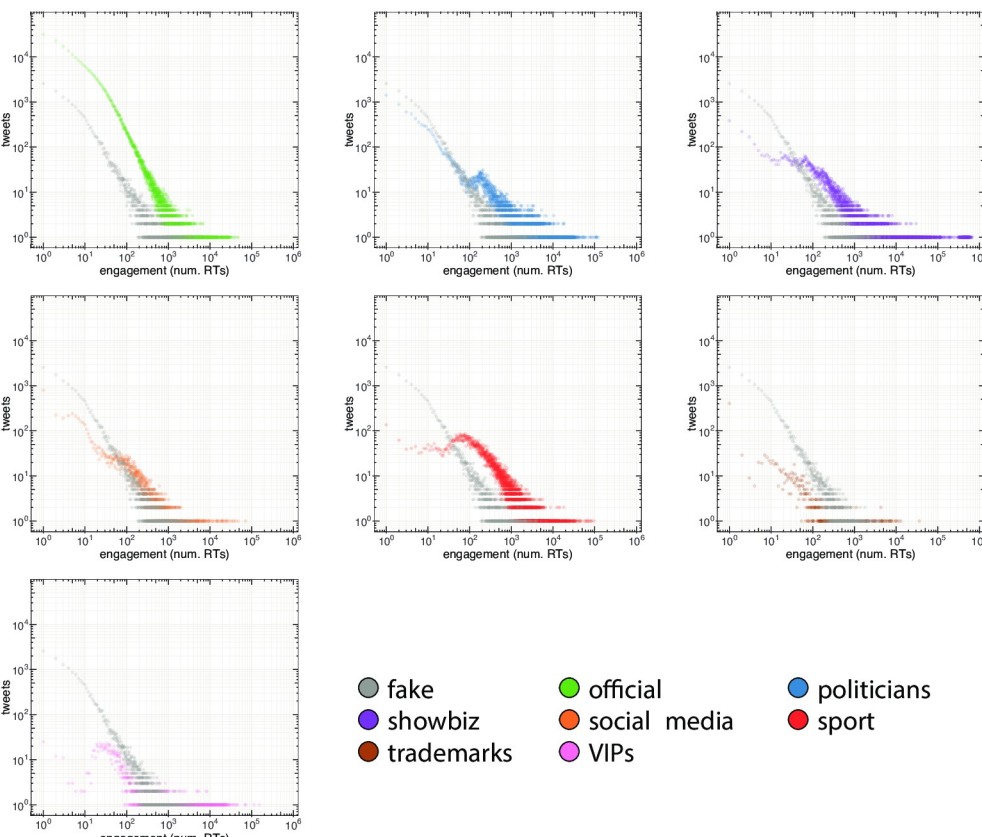

**Fig 5. Distribution of the engagement obtained by tweets of disinformation outlets (grey-colored) and comparison with the engagement obtained by tweets of all other classes.** Overall, disinformation outlets obtain less engagement than others, as shown by their distribution spanning smaller values on the *x* axis. Engagement for a given tweet is computed as the number of retweets obtained by that tweet.

tweet obtains a certain engagement that derives from the interactions (e.g., retweets) of other users with that tweet. We can thus aggregate the engagement obtained by all tweets of a given actor, to have an indication of the engagement obtained by that actor. Similarly, we can aggregate the engagement obtained by all tweets from actors of a given class (e.g., politicians, fake news outlets, etc.), to have an indication of the engagement obtained by that class of actors. In our study, engagement obtained by a given tweet is simply computed as the number of retweets that tweet obtained. With respect to other measures of engagement (e.g., the number of likes/favorites to a tweet), retweets provide an indication for how much a message spread. As such, they arguably represent a good indicator for investigating the *reach* of fake and authoritative news, which is the goal of our study. Fig 5 compares the distribution of the engagement generated by all tweets of disinformation outlets (grey-colored), with those generated by tweets of all the other classes. Overall, Fig 5 shows that the engagement obtained by disinformation outlets is lower than that obtained by all other classes. In other words, tweets from accounts in the fake class, tend to receive less retweets than those obtained by other accounts. To dig deeper into this issue, we also considered the popularity of the accounts belonging to the different classes of actors. As a measure of popularity for an account, we considered its number of followers. Then, we compared the relation between the popularity of our accounts and the mean engagement they obtain, for the different classes of actors. Results are shown in Fig 6 by means of a bi-dimensional kernel density estimation, for the 6 biggest

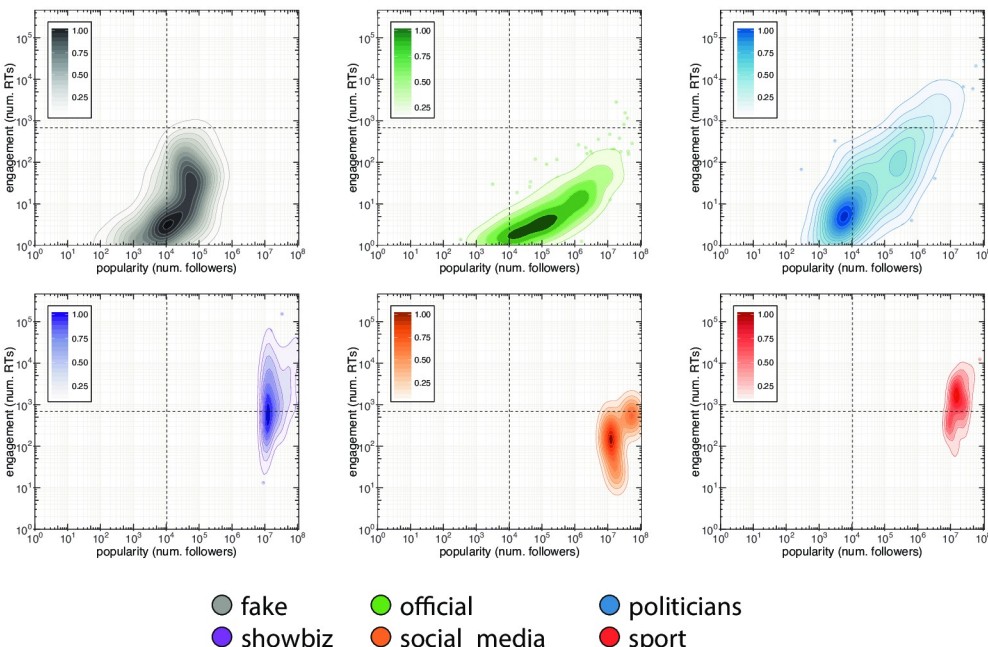

**Fig 6. Kernel density estimation of engagement and popularity of the accounts belonging to the main classes of actors.** Despite obtaining overall less engagement, disinformation outlets (grey-colored) actually obtain more engagement than official news outlets (green-colored) at middle and low popularity levels. Popularity for a given user is computed as its number of followers. Engagement for a given user is computed as the mean number of retweets obtained by tweets of that user.

classes of actors. When we consider also the popularity of the accounts, an important feature of disinformation outlets emerges. Indeed, for mid-low levels of popularity (number of followers ≤ 100,000) accounts linked to the spread of disinformation actually obtain more engagement than official news outlets, and almost the same engagement obtained by politicians. This finding is also shown in Fig 7, where popularity is logarithmically quantized into 7

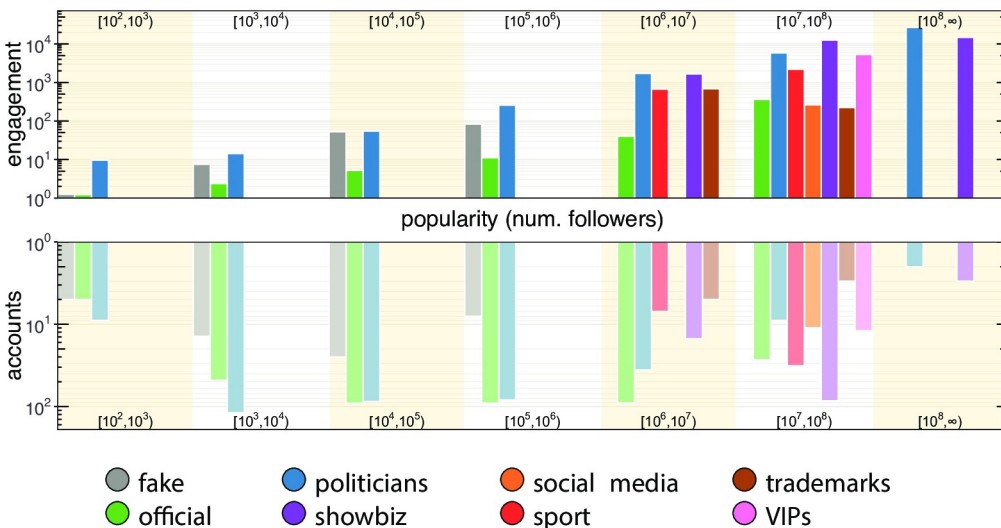

**Fig 7. Engagement obtained at different popularity levels by the different classes of actors.** Although disinformation outlets (labeled `fake`) do not reach high popularity levels, they consistently obtain more engagement than official news outlets at middle and low popularity levels, and comparable engagement with respect to politicians.

buckets. This important finding suggests that the audience of disinformation accounts is more active and more prone to share contents, with respect to that of the other classes. Anyway, no disinformation outlet currently reaches high levels of popularity (number of followers ≥1M), in contrast with all other classes of actors. As a consequence, highly popular official news outlets still obtain more engagement than disinformation outlets. This indicates that, although the audience of disinformation outlets is more prone to share information than others, their influence on the public debate remains rather limited. Additionally, even though disinformation accounts make efforts to attract interest of other central users, they cannot really fit into the information flow in any significant way.

## Conclusions

We analyzed the interactions of several accounts belonging to different figures of the public society in the context of the 2019 European Parliament elections. To have a wide view of the phenomenon, we included in our dataset also personalities not directly related to politics, such as show business and sport figures, together with a set of disinformation outlets. We collected all the tweets made by the selected accounts in the three months before the election days and we performed a quantitative analysis to identify the characteristics of the debate. By leveraging a semi-automated geolocalization technique, we also performed a geographical analysis of the phenomenon. Results show that the debate on Twitter rarely crossed national borders—that is, accounts tended to interact mainly with others coming from the same nation. Moreover, there was a strong tendency of intra-class interaction—that is, accounts mainly mentioned others from the same class. The only relevant exception were accounts of official news outlets, especially those located in the United Kingdom, that had a non-negligible percentage of links pointing to the US. Moreover, it is interesting that disinformation outlets did not interact among themselves, but rather they exhibited a tendency towards self-mentions and they tried to catch the attention of other popular accounts. Nevertheless, differently from official news outlets, disinformation outlets were almost completely ignored by other actors, thus holding a peripheral position in the interaction network and having a limited influence on the information flow. Still, they exhibited an outreach on general public higher than official news outlet and comparable with the politicians at the same levels of popularity, thus implying that the user base of disinformation outlets was more active than that of other classes of actors. However, all other categories overcame disinformation outlets in terms of absolute maximum outreach, thanks to their significantly larger absolute popularity. Finally, the limited and bounded contribution that disinformation outlets had on the overall interactions suggests that the strategies employed by Twitter to counteract the spreading of disinformation—that is, the ban or suspension of suspicious accounts—may have had a mitigation effect on the spreading of fake news thus preserving the integrity of the Twittersphere.

## Materials and methods

### Data collection

Our dataset is based on a list of 863 Twitter accounts, split across 8 categories and 18 countries. A pseudonymized version of our dataset is publicly available on GitHub (https://github.com/cinhelli/Limited-Reach-Fake-News-Twitter-2019-EU-Elections). Initially, we only considered in our study the 5 biggest European countries (UK, Germany, France, Italy and Spain) and the US. Then, other countries were added when we extended the dataset to also include popular users that interacted with users from our initial set.

The first category of accounts (labeled fake) in our study is related to known disinformation outlets. It contains 49 Twitter accounts responsible for sharing disinformation, identified

in authoritative reports—such as Reuters' Digital News Report 2018 [31] and a report from the European Journalism Observatory [32]—and fact-checking Web sites—such as Snopes [33] and Politifact [34]. Our list of official news outlets (labeled `official`) contains 347 Twitter accounts. It includes accounts corresponding to the main news outlets in each of the considered countries, derived from the media list released by the European Media Monitor [35], as well as the Twitter accounts of the main US news outlets. We then considered a list of 349 politicians (labeled `politicians`). This list includes all available Twitter accounts of the members of the European parliament [36] as of March 2019, as well as the main politicians for each considered country that did not belong to the European parliament.

We firstly exploited Twitter APIs to crawl the timelines of all the accounts belonging to the 3 previous lists. In order to match the electoral period, we only retained tweets shared between February 28 and May 22, 2019. After this step, we also manually classified a small subset of popular users (more than 1M followers) that interacted with those of our initial list in the considered time period. These accounts were classified in 5 additional categories, based on their role in the society. In this way, we obtained additional 100 `showbiz` accounts (e.g., actors, tv hosts, singers), 10 `social media` accounts (e.g., Youtube's official account), 37 `sport` accounts (e.g., sport players and the official accounts of renown sport teams), 6 `trademarks` accounts related to famous brands (e.g., Nike, Adidas) and 11 accounts of `VIPs` (e.g., the Pope, Elon Musk, J.K. Rowling). For each of these additional accounts, we crawled the respective timeline and only retained tweets shared in our considered time period.

After this data collection process, we ended up with the dataset summarized in Table 1, comprising more than 850 labeled accounts and almost 400,000 tweets.

## Account interactions

For each account, we also computed its interactions with other accounts. In particular, we split interactions into 4 different categories: retweets, replies, mentions, and article mentions.

The first 3 types of interactions are straightforward, while an article mention is detected when an account shares a tweet containing a URL to a Web page that mentions one of the labeled accounts in our dataset. To obtain information about article mentions we scraped all the Web pages linked within the tweets of our dataset. Within each page, we performed language detection and named entity recognition. Finally, we cross-checked person named entities with our lists of users.

## Account geolocation

Whenever possible, we also exploited the *location* field of Twitter accounts (both the 863 labeled ones, as well as all others with which they interacted) in order to geolocate them.

For this process, we exploited several different geolocators (e.g., Google Maps, Bing, Geo-Names) that offer their services via Web APIs. We first selected all accounts with a non-empty *location* field. Then, we built a blacklist for discarding those locations that were too vague or clearly ironic (e.g., global, worldwide, Mars, the internet), as is frequently the case with user-generated input. For each distinct location that was not removed during the filtering step, we queried one of the available geolocators and we associated the corresponding geographic coordinates to all accounts with that location.

## Ethics statement

The whole data collection process was carried out exclusively via the official Twitter APIs, which are publicly available, and for the analysis we only used publicly available data (users with privacy restrictions are not included in our dataset). The sources (i.e., accounts and Web

pages) from which we downloaded data are all publicly available. User content contributing to such sources is also public, unless the user's privacy settings specify otherwise, in which case data would not be available to us. We complied with the terms, conditions, and privacy policies of the respective websites (e.g., Twitter) and with the EU GDPR (General Data Protection Regulation, https://gdpr-info.eu/).

## Supporting information

**S1 File.**
(PDF)

## Author Contributions

**Conceptualization:** Matteo Cinelli, Stefano Cresci, Alessandro Galeazzi, Walter Quattrociocchi, Maurizio Tesconi.

**Data curation:** Matteo Cinelli, Stefano Cresci, Alessandro Galeazzi, Maurizio Tesconi.

**Formal analysis:** Matteo Cinelli, Stefano Cresci, Alessandro Galeazzi.

**Investigation:** Matteo Cinelli, Stefano Cresci, Alessandro Galeazzi.

**Methodology:** Matteo Cinelli, Stefano Cresci, Alessandro Galeazzi, Walter Quattrociocchi, Maurizio Tesconi.

**Visualization:** Matteo Cinelli, Stefano Cresci, Alessandro Galeazzi.

**Writing – original draft:** Matteo Cinelli, Stefano Cresci, Alessandro Galeazzi, Walter Quattrociocchi, Maurizio Tesconi.

**Writing – review & editing:** Matteo Cinelli, Stefano Cresci, Alessandro Galeazzi, Walter Quattrociocchi, Maurizio Tesconi.

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
