## [Decision Letter · Decision Letter 0]

13 Mar 2020

PONE-D-19-32938

The Limited Reach of Fake News on Twitter during 2019 European Elections

PLOS ONE

Dear Mr Galeazzi,

Thank you for submitting your manuscript to PLOS ONE. After careful consideration, we feel that it has merit but does not fully meet PLOS ONE’s publication criteria as it currently stands. Therefore, we invite you to submit a revised version of the manuscript that addresses the points raised during the review process.

Although both reviewers recognize the potential importance of this study, they told that the analysis you conducted is not fully valid from technical and statistical viewpoints. I agree with their opinions and I found that the points they pointed out could be crucial. Please revise your manuscript according to their suggestions.

We would appreciate receiving your revised manuscript by Apr 27 2020 11:59PM. To enhance the reproducibility of your results, we recommend that if applicable you deposit your laboratory protocols in protocols.io, where a protocol can be assigned its own identifier (DOI) such that it can be cited independently in the future. For instructions see: http://journals.plos.org/plosone/s/submission-guidelines#loc-laboratory-protocols

We look forward to receiving your revised manuscript.

Kind regards,

Yohsuke Murase, Ph.D.

Academic Editor

PLOS ONE

Journal Requirements:

"MC acknowledges the support from CNR-PNR National Project DFM.AD004.027 289

"Crisis-Lab" and P0000326 project AMOFI (Analysis and Models OF social medIa).".

i) We note that you have provided funding information that is not currently declared in your Funding Statement. However, funding information should not appear in the Acknowledgments section or other areas of your manuscript. We will only publish funding information present in the Funding Statement section of the online submission form.

ii) Please remove any funding-related text from the manuscript and let us know how you would like to update your Funding Statement. Currently, your Funding Statement reads as follows:

"he authors received no specific funding for this work".

3.  Your ethics statement must appear in the Methods section of your manuscript. If your ethics statement is written in any section besides the Methods, please move it to the Methods section and delete it from any other section. Please also ensure that your ethics statement is included in your manuscript, as the ethics section of your online submission will not be published alongside your manuscript.

Reviewers' comments:

Reviewer's Responses to Questions

**Comments to the Author**

1. Is the manuscript technically sound, and do the data support the conclusions?

Reviewer #1: Partly

Reviewer #2: Partly

2. Has the statistical analysis been performed appropriately and rigorously? 

Reviewer #1: Yes

Reviewer #2: No

3. Have the authors made all data underlying the findings in their manuscript fully available?

Reviewer #1: Yes

Reviewer #2: No

4. Is the manuscript presented in an intelligible fashion and written in standard English?

Reviewer #1: Yes

Reviewer #2: Yes

5. Review Comments to the Author

Reviewer #1: This is a descriptive paper about Twitter data analysis related to the 2019 European Parliament election. The authors analyzed approximately 400,000 tweets collected from 863 accounts including fake news and official media sources. They concluded that fake news accounts did not have a major influence on political discussions on Twitter during the election. Although this kind of case study is important for investigating online disinformation, I have several concerns in regard to the current manuscript.

1. I am not sure whether the 2019 European Parliament election is a proper case for the authors’ purpose. Are there instances of known fake news or propaganda that was spread online and became a serious issue during the 2019 European Parliament election like those that emerged during Brexit and the US election in 2016? I am concerned that this election might represent an exceptional case.

2. The second comment is related to the previous one. The current manuscript overstates the peripheral role of fake news accounts and the effectiveness of the ban on suspicious accounts. First, 49 fake news accounts in this study were depicted from the lists of authoritative reports, which is a small subset of potential malicious users. Second, the authors did not examine the effect of the ban using any quantitative methodology. They have to be careful about concluding that they have. In actuality, a large volume of malicious accounts and fake news have been reported in the case of the 2019 Hong Kong protests.

3. The authors analyzed only the number and patterns of interactions (i.e., mentions, replies, retweets, URLs) among 80 account types and did not analyze tweet texts. These are, therefore, unknown content and sentiment-level differences in fake news accounts as well as other account types. FYI, the following paper is relevant to this study:

Stella, M., Ferrara, E. & De Domenico, M. Bots increase exposure to negative and inflammatory content in online social systems. PNAS (2018).

Minor comments:

• Please place the title for Table 1 above the table.

• In L. 165, the authors referred to a previous finding, which needs pointers to the references.

Reviewer #2: This paper analyzed the impact of information (tweets) originating from 863 major accounts during the 2019 European election period together with its direct and indirect interactions between them. As a result, on the debate, the 45 fake news (=disinformation outlet) accounts had little interaction with accounts that belong to other classes such as official, even fake accounts tried to reach other accounts. However, the engagement of fake accounts with moderate popularity is stronger than other class accounts. The authors suggested that fake accounts’ influence is maintained by some limited persistent followers.

Although it is an interesting result that they quantify the interaction between fake accounts and others both in a direct and indirect way, however, there are some unclear points in method and explanations. For example, the authors identified five classes (showbiz, social media, sports, trademark vip) in accounts based on most interacted with fake, official, and politicians accounts. How did the authors define “most” interacted? Furthermore, since the result sounds similar to Stella(2018), the authors should discuss the difference.

M. Stella, E. Ferrara, and M. De Domenico, “Bots increase exposure to negative and inflammatory content in online social systems,” Proc. Natl. Acad. Sci., vol. 115 (49) pp. 12435–12440 (2018).

Fig.1:

My main concern is comparing account nationalities and classes (official, sports, etc.). I am not sure there is a significant difference between them.

For example, in a country-by-country context, Oceania accounts have 0.38% in total, perhaps only 3 accounts, which makes it difficult to discuss. In a class-by-class context, official and political have more than 300 accounts, while trademark and social media have fewer than 10. Furthermore, it’s also not clear the criteria to separate five classes.

line.95 & Fig.2

> This may be a first signal that the interactions rarely cross national borders.

I am not sure it is because of its national borders. Did the authors consider language differences? If all accounts that considered here use English, the authors should mention that, and if not, it is possible that the limited interactions occurred because of languages, too.

line.168

> the engagement, computed as the number of retweets, generated by the different classes of actors.

About the definition of the “engagement,” if the authors computed as the number of retweets generated by the different classes of actors, how did they separate each retweet by the different classes of actors? If they simply counted a number of retweets, I suggest using the word “retweets” instead of “engagement” to avoid confusion. Furthermore, did they include mention and reply?

line.175

> popularity, computed as the number of followers, of the accounts belonging to the different classes of actors.

Similar to the “engagement,” if the authors computed as the number of followers, I suggest using the word “followers” instead of “popularity” to avoid confusion.

line.172 & Fig.5

> Figure 5 shows that the engagement of disinformation outlets is dominated by that of all other classes

I do not understand Figure 5 fully. What does “dominate” mean here? The vertical axis of Figure 5 is “users,” but isn’t it a “tweet”? Since the total number of social media accounts is 8, the vertical axis of Figure 5 for orange one seems strange.

Data availability:

I checked the pointed web site and found a list of 863 accounts, but I could not find information about the interactions such as reply, mention, and others. https://github.com/cinhelli/Limited-Reach-Fake-News-Twitter-2019-EU-Elections

6. PLOS authors have the option to publish the peer review history of their article (what does this mean?). If published, this will include your full peer review and any attached files.

Reviewer #1: Yes: Kazutoshi Sasahara

Reviewer #2: No

---

## [Author Response · Author response to Decision Letter 0]

8 May 2020

All the comments made by Reviewer 1 are discussed in section 1 of "Response to Reviewer" file and

all the comments of Reviewer 2 are discussed in section 2 of "Response to Reviewer" file.

Editor: founding information have been moved from the paper to the online form section. you may update the Funding Statement to read: "MC acknowledges the support from CNR-PNR National Project DFM.AD004.027 289

"Crisis-Lab" and P0000326 project AMOFI (Analysis and Models OF social medIa)."

---

## [Decision Letter · Decision Letter 1]

2 Jun 2020

The Limited Reach of Fake News on Twitter during 2019 European Elections

PONE-D-19-32938R1

Dear Dr. Galeazzi,

We are pleased to inform you that your manuscript has been judged scientifically suitable for publication and will be formally accepted for publication once it complies with all outstanding technical requirements.

With kind regards,

Yohsuke Murase, Ph.D.

Academic Editor

PLOS ONE

Additional Editor Comments (optional):

Reviewers' comments:

Reviewer's Responses to Questions

**Comments to the Author**

1. If the authors have adequately addressed your comments raised in a previous round of review and you feel that this manuscript is now acceptable for publication, you may indicate that here to bypass the “Comments to the Author” section, enter your conflict of interest statement in the “Confidential to Editor” section, and submit your "Accept" recommendation.

Reviewer #1: All comments have been addressed

Reviewer #2: All comments have been addressed

2. Is the manuscript technically sound, and do the data support the conclusions?

Reviewer #1: (No Response)

Reviewer #2: Yes

3. Has the statistical analysis been performed appropriately and rigorously? 

Reviewer #1: (No Response)

Reviewer #2: Yes

4. Have the authors made all data underlying the findings in their manuscript fully available?

Reviewer #1: Yes

Reviewer #2: Yes

5. Is the manuscript presented in an intelligible fashion and written in standard English?

Reviewer #1: Yes

Reviewer #2: Yes

6. Review Comments to the Author

Reviewer #1: In the revised version, the authors have added a background of this study and also made reference to Stella et al. (2019). These changes would help readers understand this paper in a right context. I think adding content analysis would be beneficial, but this is not mandatory. The current manuscript could be a useful case report on social interaction patterns during 2019 European elections.

Reviewer #2: Even I do not understand the comment from the authors, "SCRIVIAMOLO ANCHE NEL PAPER MAGARI", the authors seemed to address all my comments.

7. PLOS authors have the option to publish the peer review history of their article (what does this mean?). If published, this will include your full peer review and any attached files.

Reviewer #1: Yes: Kazutoshi Sasahara

Reviewer #2: No

---

## [Editor Report · Acceptance letter]

8 Jun 2020

PONE-D-19-32938R1 

The Limited Reach of Fake News on Twitter during 2019 European Elections 

Dear Dr. Galeazzi:

I'm pleased to inform you that your manuscript has been deemed suitable for publication in PLOS ONE. Congratulations! Your manuscript is now with our production department. 

Kind regards, 

on behalf of

Dr. Yohsuke Murase 

Academic Editor

PLOS ONE